# Treatment profiles and clinical outcomes of COVID-19 patients at private hospital in Jakarta

Diana Laila Ramatillah 📍 *, Suri Isnaini

Faculty of Pharmacy, Universitas 17 Agustus 1945, Jakarta, Indonesia

* diana.ramatillah@uta45jakarta.ac.id, dianalailaramatillah@gmail.com

## Abstract

### Background

Severe Acute Respiratory Syndrome Coronavirus-2 (SARS-CoV-2) is a virus that causes COVID-19, which has become a worldwide pandemic. However, until now, there is no vaccine or specific drug to prevent or treat COVID-19.

### Objectives

To find out the effective treatment as an antiviral agent for COVID-19, to determine the correlation between sociodemography with clinical outcomes and duration of treatment, and to determine the relationship between comorbidities with clinical outcomes and duration of treatment for COVID-19 patients.

### Methods

A prospective cohort study was conducted in this study. This study included only confirmed COVID-19 patients who were admitted to the hospital during April-May 2020. Convenience sampling was used to select 103 patients, but only 72 patients were suitable for inclusion.

### Results

The survival analysis for COVID-19 patients using the Kaplan Meier method showed that patients receiving Oseltamivir + Hydroxychloroquine had an average survival rate of about 83% after undergoing treatment of about ten days. Gender (p = 0.450) and age (p = 0.226) did not have a significant correlation with the duration of treatment for COVID-19 patients. Gender (p = 0.174) and age (p = 0.065) also did not have a significant correlation with clinical outcome of COVID-19 patients. Comorbidities showed a significant correlation with duration of treatment (p = 0.002) and clinical outcome (p = 0.014) of COVID-19 patients.

### Conclusion

The most effective antiviral agent in this study based on treatment duration was the combination of Oseltamivir + Hydroxychloroquine. The higher the patient's average treatment duration is, the lower the average survival rate for COVID-19 patients.

**Data Availability Statement:** All relevant data are within the manuscript and its Supporting Information files.

**Funding:** The author(s) received no specific funding for this work.

**Competing interests:** The authors have declared that no competing interests exist.

## Introduction

Coronaviruses (CoVs) are a part of the Coronaviridae family [1]. They spread among multiple hosts, clinically presenting various symptoms, such as common cold-like to severe, sometimes deadly, respiratory infections [1]. The new virus, which is responsible for this outbreak, was initially referred to as "2019-nCoV" or "SARS-CoV-2" [1]. The pathogen that causes Severe Acute Respiratory Syndrome (SARS), SARS-CoV, is believed to be familiar with SARS-CoV-2 [1]. SARS-CoV-2 was recently closely related to SARS-CoV, which has 80% identity in the RNA sequence [1–3].

COVID-19 (Coronavirus Diseases 2019) outbreak has been here since December 2019 [1, 4]. COVID-19 ranges from mild self-limiting respiratory disorders to severe progressive pneumonia, which causes multiple organ failure and death [1, 4]. The pandemic's epicenter was Wuhan City in the province of Hubei, central China [1]. One of the first locations where SARS-CoV-2 could cross the species barrier at the animal-human interface was the Huanan seafood market [1]. Initial research conducted in Shenzhen provided the first evidence that SARS-CoV-2 can be transmitted from human-to-human [1, 5].

In a study of 44,672 individuals (1,023 deaths), the Chinese Centers for Disease Control and Prevention reported that cardiovascular disease, hypertension, diabetes, respiratory disease, and cancer were associated with an increased risk of death [6]. Advanced age, comorbidities, particularly hypertension, diabetes, obesity, and smoking, are factors that raise the risk of presenting severe diseases. [7]. However, each country's demographic and epidemiological profiles may affect the characteristics of COVID-19 [1]. In Indonesia, 51.9% of confirmed cases are men, 31.4% of confirmed cases are aged between 31–45 years, with the highest percentage of deaths between 46–59 years of age at 39.4%, and 50.5% of confirmed cases had hypertension as a comorbid disease [8]. To determine the suitability of mitigation strategies and to set goals for managing the COVID-19 pandemic, it is crucial to assess the instantaneous mortality rate at any time during monitoring for specific risk factors [9]. To overcome Covid 19, currently, countries focus on tackling an epidemic and suppressing the spreading [1].

The pandemic of COVID-19 involves the rapid creation of a useful therapeutic approach in which three principles are used, which are: (i) Testing currently known antiviral agents and verifying their clinical utility [1, 10, 11]. (ii) Enabling high computing power and simultaneous verification of millions of potential agents [1, 11, 12]. (iii) Targeted therapy, which is intended to disrupt the viral genome and function. Properly particles are designed to interfere with crucial viral infection steps, such as binding to cell surfaces and internalization [1]. Unfortunately, *in vitro* activity does not always show the same result as *in vivo* testing due to different pharmacodynamic and pharmacokinetic properties [1, 11, 13]. Antiviral drugs, selected antibiotics, antimalarials, and immunotherapeutic drugs can be useful in treating COVID-19 [1].

## Materials and methods

### Study design and setting

The research was carried out at a private hospital in Jakarta, Indonesia. The study used a prospective and retrospective cohort design, included 103 COVID-19 patients, but only 72 patients were suitable for inclusion criteria. All COVID-19 patients who got favipiravir and/or oseltamivir and / or chloroquine and / or hydroxychloroquine were included in the study. Patients having comorbid HIV/AIDS or cancer and pregnant patients were excluded from the study.

### Ethical approval

Ethical approval was sourced from the ethical medical committee from the Faculty of Health in Indonesia, and an approval letter, NO:0303–20.283/DPKE-KEP/FINAL-EA/UEU/IX/2020, was given before data collection.

### Data collecting and handling

Based on Fig 1, ethical approval was the requirement before conducting this study. The researcher would define the patients by the list of patients in the ward. Before taking the medical record data, the researcher would explain the research and its purpose to the patients with the staff's help. The informed consent was signed as an agreement of the study from the patients. The data were arranged according to socio-demography status and current medication and transferred to clinical research form (CRF). Data were analyzed descriptively by Chi-Square and Kaplan Meier test using SPSS 22 version software. Significance correlation was showed by P-value < 0.05.

## Results and discussion

### Correlation between socio-demography and duration of treatment of COVID-19 patients

Based on **Table 1,** based on gender, most of the patients were male, 45 people (62.5%). In Indonesia, as of August 6, 2020, the number of male patients with confirmed COVID-19 was 52.1% and women 47.9% [8]. There is no clear trend in which COVID-19 is more likely to be diagnosed. A confirmed diagnosis indicates that there have been laboratory tests done. In other words, there is approximately the same number of cases among men and women worldwide. There is no proof from this national survey data that men are more likely than women to contract it [14].

Based on age, most patients were in the 39–58 years age group, as many as 32 people (44.4%), and the least number in the 79–85 years age group was two people (2.8%). In Indonesia, as of August 6, 2020, the highest number of confirmed COVID-19 patients was in the 31–45 year age group, as much as 31.4% [8]. The median age of COVID-19 patients was 56 years, ranging from 18–87 years, and most patients were male [15]. Whereas in Guan et al., the patients' median age was 47 years, and 41.9% of patients were women [16].

Sig. Value of gender and age showed a value > 0.05, which means that gender and age did not correlate with duration of treatment of COVID-19 patients. There are still few studies

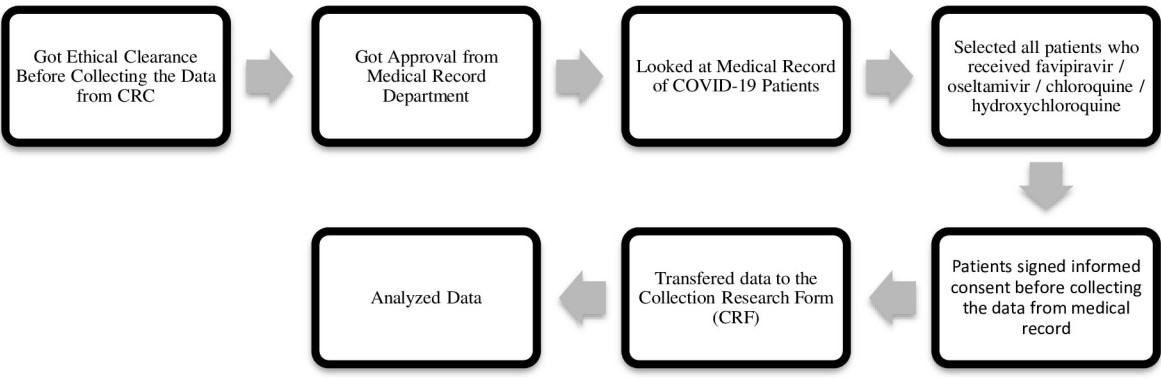

**Fig 1. Research framework of the study.**

**Table 1. Correlation between socio-demography and duration of treatment.**

| Indicators | Duration of Treatment | | Total | Sig. (*P-Value*) |
|---|---|---|---|---|
| | ≤ 14 days | > 14 days | | |
| | n (%) | n (%) | | |
| **Gender** | | | | |
| a. Male | 30 (60,0) | 15 (40,0) | 45 | 0,450 |
| b. Female | 22 (55,5) | 12 (45,5) | 27 | |
| **Age** | | | | |
| a. 19–38 years | 6 (37,5) | 10 (62,5) | 16 | 0,226 |
| b. 39–58 years | 22 (68,8) | 10 (31,2) | 32 | |
| c. 59–78 years | 13 (59,1) | 9 (40,9) | 22 | |
| d. 79–85 years | 1 (50,0) | 1 (50,0) | 2 | |

*Chi-square* test.

investigating the period of hospitalized COVID-19 patients during the pandemic. The mean duration of stay due to COVID-19 has been reported in several studies in China, 10–13 days [16, 17]. However, the duration of stay depends on different factors, such as the period elapsed from the time of exposure to the onset of symptoms, the time of onset to the time of admission to the hospital, and various aspects of the country-specific context [18].

## Correlation between socio-demography and clinical outcome of COVID-19 patients

Table 2 showed that most 45 male patients get clinical outcomes healed; as many as 30 people (66, 7%) and 15 people (33.3%) died. The 27 female patients got clinical outcomes, 22 people (81.5%) healed, and 5 people (18.5%) died. Gender does not correlate with the clinical outcome of COVID-19 patients with a P-value 0.174 > 0.05.

Furthermore, it is known that out of 16 patients aged 19–38 years, 14 people (87.5%) got clinical outcomes healed. From 32 patients aged 39–58 years, 23 people (71, 8%) got clinical outcomes recovered. Patients aged 59–78 years got clinical outcomes healed, as many as 15 people (59.1%), and the two patients aged 79–85 all (100%) died. Age does not correlate with the clinical outcome of COVID-19 patients with a P-value 0.065 > 0.05.

**Table 2. Correlation between socio-demography and clinical outcome.**

| Indicators | Clinical Outcome | | Total | Sig. (*P-Value*) |
|---|---|---|---|---|
| | Healed | Death | | |
| | n (%) | n (%) | | |
| **Gender** | | | | |
| a. Male | 30 (66,7) | 15 (33,3) | 45 | 0,174 |
| b. Female | 22 (81,5) | 5 (18,5) | 27 | |
| **Age** | | | | |
| a. 19–38 years | 14 (87,5) | 2 (12,5) | 16 | 0,065 |
| b. 39–58 years | 23 (71,8) | 9 (28,2) | 32 | |
| c. 59–78 years | 15 (68,2) | 7 (31,8) | 22 | |
| d. 79–85 years | 0 (0,0) | 2 (100,0) | 2 | |

*Chi-square* test.

**Table 3. Correlation between comorbidities and duration of treatment.**

| Comorbidities | Duration of Treatment | | Total | Sig. (*P-Value*) |
|---|---|---|---|---|
| | ≤ 14 days | > 14 days | | |
| | n (%) | n (%) | | |
| a. Without Commorbid | 6 (85,7) | 1 (14,3) | 7 | 0,002 |
| b. Pneumonia | 1 (10,0) | 9 (90,0) | 10 | |
| c. Pneumonia–Degenerative diseases | 35 (63,6) | 20 (36,4) | 55 | |

*Chi-square* test.

An increased risk of death is associated with older age, higher SOFA scores, and d-dimers greater than 1 μg/mL [15]. There were more male than female patients infected with COVID-19 [19]. More men than women were also infected with MERS-CoV and SARS-CoV [12, 20]. The decrease in women's vulnerability to viral infections can be due to the defense of the X chromosome and sexual hormones, which play a crucial role in innate and adaptive immunity [21]. Chen et al. found that, due to these patients' weaker immune function, SARS-CoV-2 was more likely to infect older adult men with chronic comorbidity [19].

## Correlation between comorbidities and duration of treatment of COVID-19 patients

Based on **Table 3** it is known that seven patients without comorbidities had a length of stay of fewer ≤ than 14 days. Meanwhile, almost most of the patients with pneumonia had a length of stay of more than 14 days (90%). Patients with pneumonia-degenerative diseases comorbidities had the largest number of patients, and 63.6% had a treatment duration of fewer than 14 days. Comorbidities correlate with the course of treatment for COVID-19 patients with a P-value of 0.002 <0.05. Based on Wang et al., The median duration from first symptoms to dyspnea, hospital admission, and ARDS was five days, seven days, and eight days, respectively [17].

## Correlation between comorbidities and clinical outcome of COVID-19 patients

Based on **Table 4** it is known that of the 7 patients who did not have comorbidities, all (100.0%) had clinical outcomes cured. Likewise, with patients who had comorbidities with pneumonia, out of 10 people (100.0%) had a clinical result healed. Patients who had comorbidities with most of the pneumonia—degenerative diseases received clinical outcomes; as many as 35 people (63.6%) and the remaining 20 people (36.4%) died. Sig value showed a value of 0.014 < 0.05, which means that comorbidities correlate with the clinical outcome of COVID-19 patients.

**Table 4. Correlation between comorbidities and clinical outcome.**

| Comorbidities | Clinical Outcome | | Total | Sig. (*P-Value*) |
|---|---|---|---|---|
| | Healed | Death | | |
| | n (%) | n (%) | | |
| a.Without Commorbid | 7 (100,0) | 0 (0,0) | 7 | 0,014 |
| b. Pneumonia | 10 (100,0) | 0 (0,0) | 10 | |
| c. Pneumonia–Degenerative diseases | 35 (63,6) | 20 (36,4) | 55 | |

*Chi-square* test.

Several reports indicate that serious, often fatal, pneumonia may be caused by COVID-19 [19, 22]. Besides, approximately half of SARS-CoV-2 infected patients had chronic underlying diseases, particularly cardiovascular and cerebrovascular diseases and diabetes, similar to MERS-CoV [20]. Cardiovascular disease contributed to more than 20% mortality cases among Covid-19 patients [23, 24]. There was a substantially higher risk of death at any point during follow-up in men, people in the older age group with chronic kidney disease, and people who were hospitalized [9].

The Chinese Centers for Disease Control and Prevention reported an increased risk of death was associated with hypertension, diabetes, cardiovascular disease, respiratory disease, and cancer [6]. Unfortunately, correction for association with age was not possible [25]. Meanwhile, a UK study showed that patients with cardiac, pulmonary, and kidney disease, cancer, dementia, and obesity had a higher risk of death [26].

## Overview of the effectiveness of COVID-19 antiviral agents

Based on **Table 5** it can be seen that from 16 patients who received Oseltamivir therapy, most of them got clinical outcomes healed (81.3%). From 28 patients who received combination therapy of Oseltamivir + Chloroquine, 16 people (57.1%) got clinical outcome cured. Four patients (50%) had healed, and four patients (50%) died from patients who received combination therapy of Oseltamivir + Hydroxychloroquine. Of the eight patients who received the combination of Favipiravir + Chloroquine therapy, all (100%) had a clinical outcome, healed. Most patients (91.67%) who received combination therapy Favipiravir + Oseltamivir + Chloroquine had healed. It was known that antiviral agent therapy correlated with the clinical outcome of COVID-19 patients (P = 0.025).

For the treatment of COVID-19, the Indonesian Society of Respirology recommends Oseltamivir because the medication is readily available in Indonesia and has been manufactured domestically [27]. In a previous study, a low-dose combination of favipiravir and oseltamivir demonstrated a synergistic response in white mice to influenza virus infection [28]. The findings of a study led by the China Pneumonia Research Network indicate that in the treatment of serious influenza, favipiravir combined with oseltamivir is better than oseltamivir alone [22]. Oseltamivir has also been used in clinical trials in various combinations with Chloroquine and favipiravir, a nucleoside analog known as a broad-spectrum antiviral drug that has shown $EC_{50}$ 61.88 μM against SARS-CoV-2 and low toxicity ($CC_{50}$>400 μM) [29].

No particular treatment for coronavirus infection has been prescribed until now, except for careful supportive care [30]. Regulation of the source of infection, personal protective procedures to minimize the risk of transmission, and early identification, isolation, and supportive treatment for infected patients are the solution to this disease. There are not adequate antibacterial agents.

**Table 5. Overview of the effectiveness of COVID-19 antiviral agents.**

| Regimen | Clinical Outcome | | Total | Sig. (P-Value) |
|---|---|---|---|---|
| | Healed | Death | | |
| | n (%) | n (%) | | |
| Oseltamivir | 13 (81,3) | 3 (18,7) | 16 | 0,025 |
| Oseltamivir + Klorokuin | 16 (57,1) | 12 (42,9) | 28 | |
| Oseltamivir + Hidroksiklorokuin | 4 (50,0) | 4 (50,0) | 8 | |
| Favipiravir + Klorokuin | 8 (100,0) | 0 (0,0) | 8 | |
| Favipiravir + Oseltamivir +Klorokuin | 11 (91,67) | 1 (8,33) | 12 | |

*Chi-square* test.

Also, there are no antiviral agents useful for the treatment of SARS and MERS [22]. All patients in the study received antibacterial agents, 90% received antiviral therapy, and 45% received methyl-prednisolone. Depending on the seriousness of the condition, the doses of oseltamivir and meth-ylprednisolone differ. However, there were no significant findings observed [29].

For a long time ago, Chloroquine (CQ) and its hydroxyl analog, Hydroxychloroquine (HCQ), have been using as antimalarial agents. Besides that, many studies have explored these drugs for the possibility of medication activity for other infectious diseases [31]. Chloroquine's mechanism for viruses is Chloroquine causes coating inhibition and/or changes in post-trans-lational modification of newly synthesized proteins in viruses, especially inhibition of glycosyl-ation [32]. Chloroquine interfering with terminal glycosylation of the cellular receptor, angiotensin-converting-enzyme-2 (ACE2), effectively prevented the spread of SARS-CoV in cell culture, which lead to Chloroquine or Hydroxychloroquine may have indirect antiviral effects [33]. Components of SARS-CoV receptors and orthomyxoviruses, sialic acid, might be inhibited by Chloroquine or Hydroxychloroquine [34].

Based on China's study, Chloroquine's effects in vitro using Vero E6 cells infected with SARS-CoV-2 at a multiplicity of infection (MOI) 0.05. This study showed that chloroquine, with an Effective Concentration (EC) of 90 of 6.90 μM, effectively reduces viral replication, which can easily be accomplished with standard doses due to its strong penetration into tissues, including the lungs. By increasing endosomal pH and interfering with SARS-CoV cellular receptors' glyco-sylation, chloroquine is known to block viral infection. The authors also speculate that the drug's documented immunomodulatory effect could improve the antiviral effect in vivo [29].

Based on Sanders et al, Oseltamivir is approved for influenza treatment, but there is no proof to fight SARS-CoV-2 in vitro activity [35]. Most patients received oseltamivir therapy during the initial COVID-19 outbreak in China due to it was occurred during the peak influ-enza season, until it was found that SARS-CoV-2 was the cause of COVID-19 [16, 22, 29]. Oseltamivir is currently used in several clinical trials as the comparison group but not as a pro-posed therapeutic intervention [36].

Favipiravir, formerly known as T-705, is a purine nucleotide prodrug, favipiravir ribofura-nosyl-5′-triphosphate. The RNA polymerase is inhibited by the active agent of favipiravir, stopping viral replication. Most of the preclinical evidence for favipiravir is derived from the activity of influenza and Ebola; however, the agent has also demonstrated broad activity against other RNA viruses [37]. The $EC_{50}$ of favipiravir against SARS-CoV-2 in vitro in Vero E6 cells was 61.88 μM/L [29, 38]. Favipiravir first reaches infected cells by endocytosis and then transformed by phosphoryibosylation and phosphorylation to the active ribofuranosyl phosphate favipiravir [37, 38]. Antiviral activity selectively targets the RNA-dependent-RNA-polymerase (RdRp) conservative catalytic domain, which can inhibit RNA polymerization activity, disrupting Curr Pharmacol Rep's nucleotide incorporation mechanism during viral RNA replication. [37, 39]. Viruses do replication RNA to increase the number and do transi-tion mutations such as guanine replacement by adenine and cytosine by thymine or by uracil, which causes lethal mutagenesis in viral RNA [37, 39]. Randomized controlled trials found that COVID-19 patients treated with favipiravir had a higher rate of recovery (71.43%) than those treated with umifenovir (55.86%) and a substantially more extended period of fever and cough relief than those treated with umifenovir [38].

## Survival analysis among COVID-19 patients

Based on **Fig 2**. it can be seen that patients who received combination therapy of Oseltamivir + Hydroxychloroquine had an average survival rate of around 83% after undergoing treatment for around ten days. Patients receiving Oseltamivir therapy had an average survival rate of

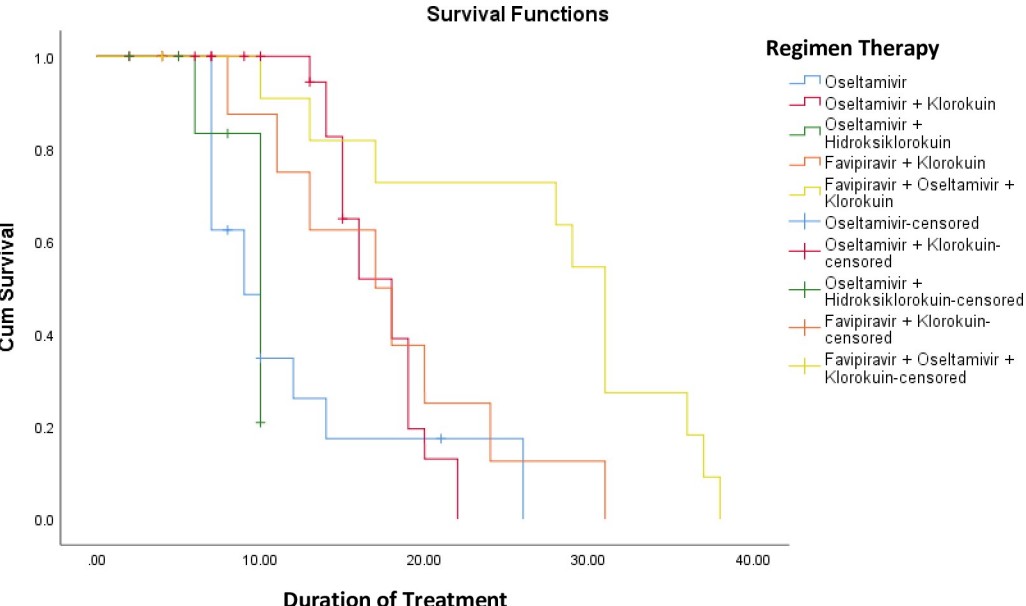

**Fig 2. Survival analysis with Kaplan Meier methods.**

approximately 18% after about 27 days of treatment. Patients receiving the combination Oseltamivir + Chloroquine therapy had an average survival rate of around 17% after around 23 days of treatment. Patients who received combination therapy Favipiravir + Chloroquine had an average survival rate of about 17% after approximately 31 days of treatment. Meanwhile, patients who received combination therapy Favipiravir + Oseltamivir + Chloroquine had an average survival rate of around 10% after undergoing treatment for about 39 days.

A significant correlation was found among the overall comparisons between survival analysis and treatment duration of COVID-19 patients. In a study conducted by Thai et al., it was shown that the mean duration of hospital stay was 21 days. The multivariable Cox regression model shows that age, residence, and contamination sources are significantly associated with extended stays in hospitals [18].

Meanwhile, in a study conducted by Wang et al., the mean duration of hospital stay was 19 days. Adjusted multivariate analysis showed that longer length of stay in hospital was associated with a factor of age 45 and more, those who were admitted to a provincial hospital, and those who were seriously ill. There was no gender difference [17].

## Conclusion

The number of COVID-19 patients who died in this study was 27.8%. The most widely used antiviral agent for patients with confirmed COVID-19 was the combination of Oseltamivir + Chloroquine. The antiviral agent therapy rated the most effective based on the duration of treatment was the combination of Oseltamivir + Hydroxychloroquine, which had the highest survival rate at around 83% after undergoing treatment for about ten days (p = 0.027). However, large sample size and multicenter study would help find the most effective antiviral agent for COVID-19.

## Limitation

The limitations that occur in this study are the relatively short research time; the extraordinary volume and speed of published literature on the treatment of COVID-19 means that research

findings and recommendations continue to evolve as new evidence emerges; and treatment data published have come exclusively from observational data or small clinical trials (none of which had more than 250 patients), which presented a higher risk of bias or inaccuracy concerning large treatment effect sizes.

## Supporting information

**S1 Data.**
(SAV)

## Acknowledgments

We acknowledge the pharmacy department's head, director of the medical record department, and all the staff who supported our research in this private hospital.

## Author Contributions

**Data curation:** Suri Isnaini.

**Formal analysis:** Diana Laila Ramatillah.

**Investigation:** Suri Isnaini.

**Methodology:** Diana Laila Ramatillah.

**Supervision:** Diana Laila Ramatillah.

**Writing – original draft:** Suri Isnaini.

**Writing – review & editing:** Diana Laila Ramatillah.

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
