## [Decision Letter · Decision Letter 0]

7 Dec 2020

PONE-D-20-31880

TREATMENT PROFILES AND CLINICAL OUTCOMES OF COVID-19 PATIENTS AT PRIVATE HOSPITAL IN JAKARTA

PLOS ONE

Dear Dr. Ramatillah,

Thank you for submitting your manuscript to PLOS ONE. After careful consideration, we feel that it has merit but does not fully meet PLOS ONE’s publication criteria as it currently stands. Therefore, we invite you to submit a revised version of the manuscript that addresses the points raised during the review process.

The reviewers have commented on your above paper. They have suggested that this manuscript be revised according to the reviewers suggestions and resubmitted.

We look forward to receiving your revised manuscript.

Kind regards,

Prof. Raffaele Serra, M.D., Ph.D

Academic Editor

PLOS ONE

Additional Editor Comments:

The reviewers have commented on your above paper. They have suggested that this manuscript be revised according to the reviewers suggestions and resubmitted.

Journal Requirements:

2. Thank you for submitting the above manuscript to PLOS ONE. During our internal evaluation of the manuscript, we found significant text overlap between your submission and the following previously published works:

https://www.sciencedirect.com/science/article/abs/pii/S0924857907002580?via%3Dihub

https://inphormative.com/tag/covid-19/

https://jamanetwork.com/journals/jama/fullarticle/2764727?resultClick=1

https://pubmed.ncbi.nlm.nih.gov/32395418/

https://pubmed.ncbi.nlm.nih.gov/32717568/

Please revise the manuscript to rephrase the duplicated text, cite your sources, and provide details as to how the current manuscript advances on previous work. Please note that further consideration is dependent on the submission of a manuscript that addresses these concerns about the overlap in text with published work.

4. We suggest you thoroughly copyedit your manuscript for language usage, spelling, and grammar. If you do not know anyone who can help you do this, you may wish to consider employing a professional scientific editing service.  

5. To comply with PLOS ONE submission guidelines, in your Methods section, please provide additional information regarding your statistical analyses and ensure you have included (1) the name and version of any software package used, alongside any relevant references, (2) the technical details or procedures required to reproduce the analysis . For more information on PLOS ONE's expectations for statistical reporting, please see https://journals.plos.org/plosone/s/submission-guidelines.#loc-statistical-reporting.

Reviewers' comments:

Reviewer's Responses to Questions

**Comments to the Author**

1. Is the manuscript technically sound, and do the data support the conclusions?

Reviewer #1: Yes

Reviewer #2: No

2. Has the statistical analysis been performed appropriately and rigorously? 

Reviewer #1: Yes

Reviewer #2: I Don't Know

3. Have the authors made all data underlying the findings in their manuscript fully available?

Reviewer #1: Yes

Reviewer #2: Yes

4. Is the manuscript presented in an intelligible fashion and written in standard English?

Reviewer #1: Yes

Reviewer #2: Yes

5. Review Comments to the Author

Reviewer #1: The authors aimed to find out the effective treatment as an antiviral agent for COVID-19, to determine the correlation between sociodemography with clinical outcomes and duration of treatment, and to determine the relationship between comorbidities with clinical outcomes and duration of treatment for COVID-19 patients.

The study, overall considered, is very interesting for the journal but in the discussion I would deepen a little more the issue of concomitant cardiovascular disease and outcomes. For example cite and discuss the paper by Ielapi N et al. Cardiovascular disease as a biomarker for an increased risk of COVID-19 infection and related poor prognosis. Biomark Med. 2020;14(9):713-716.

Reviewer #2: This is the first submission for an article titled “Treatment profiles and clinical outcomes of COVID-19 patients at private hospital in Jakarta”. The project has a well written introduction. I am largely concerned about the manuscript sample size and the number of analyses completed with the size of the sample. The statistical plan is also not fully articulated relative to the aims. Additionally, the style of the paper is not traditional and would need to be altered to the style of the journal and typically accepted format.

6. PLOS authors have the option to publish the peer review history of their article (what does this mean?). If published, this will include your full peer review and any attached files.

Reviewer #1: No

Reviewer #2: No

---

## [Author Response · Author response to Decision Letter 0]

8 Jan 2021

FOR EDITOR

1. We have improved the article according to the guidelines

2. We have corrected the significant text overlap (all the read colour)

3. We have provided the details regarding participant consent (in Methodology). Actually, my student, Suri Isnaini, worked as a part-time job in the hospital's pharmacy department. That’s why she could get the data because she was one of the staff who prepared the medication for the patients. 

4. We have corrected the spelling and the grammar from the English center of our Univ

For Reviewer

1. Is the manuscript technically sound, and do the data support the conclusions?

We have corrected the conclusion

2. Has the statistical analysis been performed appropriately and rigorously?

Yes, we just used Chi-Square and Kaplan Meier. It based on our data only

3. Reviewer #1: The authors aimed to find out the effective treatment as an antiviral agent for COVID-19, to determine the correlation between sociodemography with clinical outcomes and duration of treatment, and to determine the relationship between comorbidities with clinical outcomes and duration of treatment for COVID-19 patients.

The study, overall considered, is very interesting for the journal but in the discussion I would deepen a little more the issue of concomitant cardiovascular disease and outcomes. For example cite and discuss the paper by Ielapi N et al. Cardiovascular disease as a biomarker for an increased risk of COVID-19 infection and related poor prognosis. Biomark Med. 2020;14(9):713-716.

(we have added as a reviewer suggestion in the yellow colour)

Reviewer #2: This is the first submission for an article titled “Treatment profiles and clinical outcomes of COVID-19 patients at private hospital in Jakarta”. The project has a well written introduction. I am largely concerned about the manuscript sample size and the number of analyses completed with the size of the sample. The statistical plan is also not fully articulated relative to the aims. Additionally, the style of the paper is not traditional and would need to be altered to the style of the journal and typically accepted format.

(We have did correction for the format and style of manuscript according to the guideline and for the analysis we just do Chi-square and Kapplan Meier) the correction of methodology appears in yellow.

6th January 2021

1) Thank you for updating your data availability statement. You note that your data are available within the Supporting Information files, but no such files have been included with your submission. At this time we ask that you please upload your minimal data set as a Supporting Information file, or to a public repository such as Figshare or Dryad.

Please also ensure that when you upload your file you include separate captions for your supplementary files at the end of your manuscript.

As soon as you confirm the location of the data underlying your findings, we will be able to proceed with the review of your submission.

2) Please ensure that you refer to Figure 1 in your text as, if accepted, production will need this reference to link the reader to the figure.

3) We note that your revised manuscript still contains significant overlap in the following sections with the following sources:

Introduction, pg. 2-3:

https://bmcpublichealth.biomedcentral.com/articles/10.1186/s12889-020-09721-2

https://pubmed.ncbi.nlm.nih.gov/32717568/

Results & Discussion, pg. 7-10:

https://www.nature.com/articles/s41586-020-2521-4

https://inphormative.com/tag/emergency/

https://www.sciencedirect.com/science/article/abs/pii/S0924857907002580?via%3Dihub

https://jamanetwork.com/journals/jama/fullarticle/2764727

https://link.springer.com/article/10.1007/s40495-020-00216-7?code=8707958e-3d60-49c1-b3af-f3e1434fba3e&error=cookies_not_supported

ANSWER:

1) We had put in Fig share

2) Ok

3) We have revised in the blue colour

---

## [Decision Letter · Decision Letter 1]

15 Mar 2021

PONE-D-20-31880R1

TREATMENT PROFILES AND CLINICAL OUTCOMES OF COVID-19 PATIENTS AT PRIVATE HOSPITAL IN JAKARTA

PLOS ONE

Dear Dr. Ramatillah,

Thank you for submitting your manuscript to PLOS ONE. After careful consideration, we feel that it has merit but does not fully meet PLOS ONE’s publication criteria as it currently stands. Therefore, we invite you to submit a revised version of the manuscript that addresses the points raised during the review process.

The manuscript was improved but some statistical issues have been raised by one of the reviewer. Please revise the  manuscript accordingly.

We look forward to receiving your revised manuscript.

Kind regards,

Prof. Raffaele Serra, M.D., Ph.D

Academic Editor

PLOS ONE

Reviewers' comments:

Reviewer's Responses to Questions

**Comments to the Author**

1. If the authors have adequately addressed your comments raised in a previous round of review and you feel that this manuscript is now acceptable for publication, you may indicate that here to bypass the “Comments to the Author” section, enter your conflict of interest statement in the “Confidential to Editor” section, and submit your "Accept" recommendation.

Reviewer #1: All comments have been addressed

Reviewer #3: (No Response)

2. Is the manuscript technically sound, and do the data support the conclusions?

Reviewer #1: Yes

Reviewer #3: Partly

3. Has the statistical analysis been performed appropriately and rigorously? 

Reviewer #1: Yes

Reviewer #3: No

4. Have the authors made all data underlying the findings in their manuscript fully available?

Reviewer #1: Yes

Reviewer #3: Yes

5. Is the manuscript presented in an intelligible fashion and written in standard English?

Reviewer #1: Yes

Reviewer #3: Yes

6. Review Comments to the Author

Reviewer #1: the manuscript has beeb properly reviewed and now it is ready for publication. Well done. Looking forward to its final publication.

Reviewer #3: This manuscript is a revised version based on the comments received during the previous round of reviews. The authors are moderately receptive to the comments raised. The statistical analysis plan is pretty straightforward; several correlation analysis, and eventually a Kaplan-Meier plot comparing various regimen therapy. Without a well-thought-of regression analysis, one can't say much about the study. However, I assume authors refrained from such regression analysis due to the relatively small sample size they could actually generate. My comments are below:

1. I am still concerned with the sample size they could actually generate. It would be great for the readers, if the authors can provide a sample/size power statement based on a desired effect size which the authors wanted to pursue at the onset, and which uses the primary outcome (the survival times, with possible right-censoring).

2. In Figure 2, I could see only the K-M plots (with descriptions in the text), however, I do not see any formal "testing" and associated p-values (likely via log-rank tests), if I am not mistaken. A null hypothesis can be tested that way, and if rejected, separate 2-group tests (controlling for multiplicity) maybe conducted.

3. K-M curves for the respective groups are crossing, so the log-rank may NOT be the most powerful under proportional hazards alternatives, but we may ignore that now. All-in-all, it is not clear whether the sample size they could collect is close to the desired one for conducting this comparison. In that regard, the study can be at best considered a very pilot study. This needs to be made clear.

7. PLOS authors have the option to publish the peer review history of their article (what does this mean?). If published, this will include your full peer review and any attached files.

Reviewer #1: No

Reviewer #3: No

---

## [Author Response · Author response to Decision Letter 1]

15 Mar 2021

I have made revision as reviewers comments

---

## [Decision Letter · Decision Letter 2]

1 Apr 2021

TREATMENT PROFILES AND CLINICAL OUTCOMES OF COVID-19 PATIENTS AT PRIVATE HOSPITAL IN JAKARTA

PONE-D-20-31880R2

Dear Dr. Ramatillah,

We’re pleased to inform you that your manuscript has been judged scientifically suitable for publication and will be formally accepted for publication once it meets all outstanding technical requirements.

Kind regards,

Prof. Raffaele Serra, M.D., Ph.D

Academic Editor

PLOS ONE

Additional Editor Comments (optional):

amended manuscript is acceptable

Reviewers' comments:

Reviewer's Responses to Questions

**Comments to the Author**

1. If the authors have adequately addressed your comments raised in a previous round of review and you feel that this manuscript is now acceptable for publication, you may indicate that here to bypass the “Comments to the Author” section, enter your conflict of interest statement in the “Confidential to Editor” section, and submit your "Accept" recommendation.

Reviewer #3: All comments have been addressed

2. Is the manuscript technically sound, and do the data support the conclusions?

Reviewer #3: (No Response)

3. Has the statistical analysis been performed appropriately and rigorously? 

Reviewer #3: (No Response)

4. Have the authors made all data underlying the findings in their manuscript fully available?

Reviewer #3: (No Response)

5. Is the manuscript presented in an intelligible fashion and written in standard English?

Reviewer #3: (No Response)

6. Review Comments to the Author

Reviewer #3: (No Response)

7. PLOS authors have the option to publish the peer review history of their article (what does this mean?). If published, this will include your full peer review and any attached files.

Reviewer #3: No

---

## [Editor Report · Acceptance letter]

6 Apr 2021

PONE-D-20-31880R2 

TREATMENT PROFILES AND CLINICAL OUTCOMES OF COVID-19 PATIENTS AT PRIVATE HOSPITAL IN JAKARTA 

Dear Dr. Ramatillah:

I'm pleased to inform you that your manuscript has been deemed suitable for publication in PLOS ONE. Congratulations! Your manuscript is now with our production department. 

Kind regards, 

on behalf of

Prof. Raffaele Serra 

Academic Editor

PLOS ONE